

# Gray level co-occurrence matrix (GLCM) texture based crop classification using low altitude remote sensing platforms

Naveed Iqbal*, Rafia Mumtaz*, Uferah Shafi and Syed Mohammad Hassan Zaidi

National University of Sciences and Technology (NUST), School of Electrical Engineering and Computer Science (SEECS), Islamabad, Pakistan
* These authors contributed equally to this work.

## ABSTRACT

Crop classification in early phenological stages has been a difficult task due to spectrum similarity of different crops. For this purpose, low altitude platforms such as drones have great potential to provide high resolution optical imagery where Machine Learning (ML) applied to classify different types of crops. In this research work, crop classification is performed at different phenological stages using optical images which are obtained from drone. For this purpose, gray level co-occurrence matrix (GLCM) based features are extracted from underlying gray scale images collected by the drone. To classify the different types of crops, different ML algorithms including Random Forest (RF), Naive Bayes (NB), Neural Network (NN) and Support Vector Machine (SVM) are applied. The results showed that the ML algorithms performed much better on GLCM features as compared to gray scale images with a margin of 13.65% in overall accuracy.

## INTRODUCTION

Remote sensing is a valuable tool in evaluating, monitoring, and management of land, water, and crop resources. Satellite imagery and aerial imagery have wide applications in the field of agriculture, monitoring snow cover trends, wildfire trends, water level rising, and forestry. In agriculture, crop health monitoring, yield estimation, classification of crops based on land cover, and monitoring of droughts are some common applications of remote sensing (*Navalgund, Jayaraman & Roy, 2007*; *Seelan et al., 2003*; *Hufkens et al., 2019*; *Sivakumar et al., 2004*). Among these applications, crop classification is quite challenging due to the texture and color similarity of crops in the initial stages. For this purpose, satellite data is commonly used which provides free access to the data. The data obtained from such platforms is coarse in a resolution which makes it difficult to classify the different types of crops. Apart from a coarse resolution of satellite data, the effect of atmospheric particles and cloud cover percentage in the image, if greater than 90% will result in discarding the images as no valuable information can be extracted from the satellite for these images.

Corresponding author
Naveed Iqbal,
niqbal.msit17seecs@seecs.edu.pk

The low-cost Un-manned Aerial Vehicles (UAV) are the substitute of the satellite platforms which provide high-resolution data with flexibility in data collection. After high-resolution images acquisition, several Machine/Deep Learning (ML/DL) algorithms are applied to classify the different types of crops. A lot of applications use texture information as features that are further given as input to the ML classification algorithms. The texture features provide useful insights into the color, its spatial arrangement, and intensities.

In *Kwak & Park (2019)*, the crop classification based on texture features is performed on the data collected by a drone mounted with a multi-spectral camera. The acquired images are up-scaled to 25 cm resolution and mosaiced later to obtain a complete field of view. To extract texture features, Gray Level Co-occurrence Matrix (GLCM) at different kernel sizes is used including $3 \times 3$, $15 \times 15$, and $31 \times 31$. The mosaiced images act as an input to classification algorithms, such as Random Forest and Support Vector Machine (SVM). It is seen that using textural features obtained from larger kernel size showed improvement in classification results by 7.72% in overall accuracy rather than only using spectral information for classification.

Similarly in *Böhler, Schaepman & Kneubühler (2018)*, texture base classification of crops is performed at pixel and parcel-based level where the crops in the study are maize, bare soil, sugar beat, winter wheat, and grassland. The images are acquired by eBee UAV in four flights of 30 min each on 26 June 2015. Textural features are extracted from the obtained UAV images. Random forest algorithm is applied after extracting the texture features which obtained an overall accuracy of 86.3%.

In this study, we performed a classification of four different types of crops including wheat, soybean, rice, maize. The main objective of this research is to investigate the texture feature-based crop classification of different crops having the same spatial texture and colors. The high-resolution optical images are acquired by the drone. The multiple texture features are extracted including contrast, homogeneity, dimensionality, angular second moment, energy, and correlation. To perform classification, Support Vector Machine, Naive Bayes, Random Forest, and Neural Network are applied on the grayscale images and the texture features.

## RELATED WORK

### Crop classification traditional techniques

Over the two decades, a lot of research has been done in the agriculture domain to perform different agricultural activities, particularly in crop disease detection, crop health monitoring, crop yield estimation, crop classification, and others (*Latif, 2018*). To perform these activities; machine learning or deep learning techniques are applied to the data collected from satellite, drone or IoT sensors which are discussed in the sections below.

### Crop classification using satellite data

An analysis on crop classification and land cover is presented in *Khaliq, Peroni & Chiaberge (2018)*, in which Sentinel-2 is used to capture the multi-spectral imagery. The phonological cycle of crops is analyzed by computing the NDVI of time series spectral

imagery data. The 'Random Forest' classifier is used to classify the land cover where NDVI values are used as feature vectors. The Random Forest shows 91.1% classification accuracy i.e. predicted land cover match with the actual ground cover. In *Deng et al. (2019)*, land cover classification is performed using Random Forest as a classifier. The images are acquired from two satellites including Landsat 8 and MODIS. These images are fused based on Enhanced Spatial-Temporal and Fusion Model to generate time series-based Landsat-8 images. The data from the GF-1 satellite and Google Earth is used as supporting data for training and validation. In this research work, object base classification is used instead of pixel-based classification. The classification results show an accuracy of 94.38% on the fused data.

In *Luciani et al. (2017)*, an analysis on crop classification is presented in which Landsat-8 OLI is used to capture the multispectral imagery at a coarse spatial resolution of 30 m. The acquired images are resampled to 15 m spatial resolution using the pan-sharpening technique. The phenological profile of crops is extracted by processing NDVI of time series spectral imagery data. The phenological profile is extracted based on pixel-level and interpolation is used for the reconstruction of missing NDVI value at a particular pixel. The univariate decision tree is applied to the data where the feature vector consists of NDVI values. Results show that the univariate decision tree achieved an accuracy of 92.66%.

There are a lot of datasets that are publicly available for land classification. In *Helber et al. (2018)*, land classification is performed using the publicly available dataset 'EuroSAT' which is comprised of 27,000 labeled examples covering 10 distinctive classes. Each image patch is 64 × 64 pixels which is collected from 30 cities in European Urban Atlas. For classification, the data set is divided in the ratio of 80 to 20 which is used for training and testing respectively. Two deep learning architectures such as 'GoogLeNet' and ResNet-50 are trained on the dataset which achieved an accuracy of 98.18% and 98.57% respectively.

In *Hufkens et al. (2019)*, the health of the wheat crop is monitored using near-surface imagery captured by a smartphone. Images are collected from 50 fields by smartphone during the complete life cycle of the wheat crop. Each day, farmers captured images three times and captured images are transmitted to the cloud where the green level is assessed by green chromatic coordinates. The crop is classified as healthy or unhealthy based on the green level. Subsequently, the classification result is compared with Landsat 8 imagery in which classification of healthy and unhealthy crops is performed based on Normalized Difference Vegetation Index (NDVI) and Enhanced Vegetation Index (EVI) values. Results show that there is a small deviation between the classification results based on smartphone imagery and satellite imagery.

## Crop classification using drone data

Textural features from an image help to extract useful information from the images. In *Liu et al. (2018)*, the experimental area is selected in Minzhu Township, Daowai District, Harbin, where the variety of crops are planted. The 12 types of cropland cover in the study include rice, unripe wheat, ripe wheat, harvested wheat, soybean, corn, trees, grassland,

bare land, houses, greenhouses, and roads. The measurement and marking of Ground Control Points (GCP) are conducted on 3 August 2017 and data is collected on 4 August 2017 using a fixed-wing UAV with a Sony Digital Camera. Digital Surface Model(DSM) and Digital Orthophoto Map (DOM) are produced with the help of POS data and GCP. Texture features mean, variance, homogeneity, contrast, dissimilarity, entropy, second moment, and correlation are extracted using ENVI software for RGB and DSM bands. SVM is used to perform the classification of crops with RBF kernel. The combination of different features is performed to see the impact of each feature. By using RGB resulted in a classification accuracy of 72.94% and a combination of RGB, DSMs, Second Moment of green band, DSMs variance (27 * 27), DSMs contrast (27 * 27) achieved an accuracy of 94.5%. The results show that the hard to differentiate classes in color space became separable by adding altitude as a spatial feature where height for each tree, crop, and grass differs.

In *Hu et al. (2018)*, a hyper-spectral imaging sensor is mounted on a UAV to offer images at a higher spatial and higher spectral resolution at the same interval. The study area chosen is a field in the southern farm in Honghu city, located in China. The images are taken from the altitude of 100 m at a spatial resolution of 4 cm with 274 spectral bands. To fully utilize the potential of the spatial and spectral resolution of the image, a combination of the CNN-CRF model is proposed, to classify crops accurately. For this to work, in preprocessing phase, the Principal Component Analysis (PCA) is performed for dimensionality reduction of the data while in meantime preserving spectral information. Each patch on the image will be passed to CNN as input, to get the rule image from the PCA. The rule image, which is the output of CNN will be passed to the CRF model to generate a classification map of the output. The CNN-CRF model achieved an accuracy of 91.79% in classifying different crop types.

Image fusion between satellite and UAV can help in the classification of crops at a detailed level. In *Zhao et al. (2019)*, a fusion between Sentinel-2A satellite and images acquired from fixed-wing Agristrong UAV drone is performed to get the image at high spatial, high spectral, and high temporal resolution. For this purpose, an experimental area covering around 750 ha is selected in Harbin city, Heilongjiang province, China. The crop types in the current study include rice, soybean, corn, buckwheat, other vegetation, bareland, greenhouses, waters, houses, and roads. The images are acquired using a UAV drone for 14 September 2017 at 0.03 m resolution and Sentinel-2A images for 16 September 2017 are downloaded. The high-resolution 0.03 m images are sub-sampled at lower resolution (0.10 m, 0.50 m, 1.00 m, and 3.00 m). The fusion between UAV images at different resolutions and Sentinel-2A images is performed using Gram-Schmidt transformation (*Laben & Brower, 2000*). Random forest algorithm performed better crop classification for a fused image at 0.10 m with accuracy at 88.32%, whereas without fusion the accuracy is at 76.77% and 71.93% for UAV and Sentinel-2A images respectively.

In *Böhler, Schaepman & Kneubühler (2018)*, classification of crops is done at pixel and parcel-based level. The study area covering 170 hectares is selected in the Canton of Zurich, Switzerland. The crops in the study are maize, bare soil, sugar beat, winter wheat,

and grassland. The images are acquired by eBee UAV in four flights of 30 min each on 26 June 2015. Subsequently, the textural features are extracted from the obtained UAV images. The random forest algorithm is applied to the extracted features and crop maps are generated where the object-based classification resulted in the overall accuracy of 86.3% for the overall set of crop classes.

## Deep learning for crop classification

In *Trujillano et al. (2018)*, a deep learning network is used to classify the corn crop in the region of Peru, Brazil. The images are acquired for two locations where the first location contained corn plots, trees, river, and other crops situated in a mountainous region, where flight is conducted at 100 and 180 m respectively. The second location is a coastal area where images are acquired at an altitude of 100 m, area consists of a corn crop and some nearby houses. The multi-spectral camera mounted on the UAV acquired images in 5 different bands, at a spatial resolution of 8 cm. Photoscan tool is used to generate the mosaic of the image. The image is divided into a patch size of $28 \times 28$, covering two rows of the cornfields. The patch is labeled as corn or no corn field. Four datasets are generated from the acquired images where dataset #1 and dataset #2 covered classes with images acquired at an altitude of 100 m and 180 m. The dataset #3 merged the corn classes from different altitude flight images whereas, in dataset #4, the dataset #1 is augmented which included rotation and flipping of images. Each dataset containing $28 \times 28$ patches of images is trained using the LeNet model, in which dataset number two achieved an accuracy of 86.8% on the test set.

In *Zhou, Li & Shao (2018)*, the various types of crop classification methods are proposed using CNN and SVM algorithms. For this purpose, Yuanyang Country, in the province of Henan, China is selected as a study area where the main crops in the region are rice, peanut, and corn. The Sentinel-2A images are acquired for two dates, where all the bands data has been resampled to 10 m resolution and the resultant stack of the 26-dimensional image is generated. A ground survey is conducted in August 2017, for the labeling of different types of crops. Around 1,200 pixels are selected for training and the rests of the pixels are used for validation. The labeled pixel in the final stack image is converted to grayscale which is given as an input to the model. The CNN outperformed the SVM, where it clearly shows the deep learning-based model is better at learning the features while achieving an accuracy of 95.61% in the classification of crops. In *Sun et al. (2020)*, an application for smart home is presented. The application monitors the moisture of the soil and the value of nitrogen, phosphorous, and potassium for an indoor plant with the help of IoT sensors. The value is classified based on various levels and provides feedback to the user with help of the dashboard. The system designed is a prototype, which helps the farmers when to irrigate the crop and what ratio of the value of nutrients is suitable for the specific plant. Water content estimation in plant leaves can help in the productivity of the crops. In *Zahid et al. (2019)*, a novel approach based on machine learning is presented to estimate the health status of the plant leaves terahertz waves by measuring transmission response for four days. Each frequency point recorded is used as a feature in the study. Feature selection was carried out to discard any irrelevant feature that could result in the

wrong prediction of water content in the leaves. The support vector machine (SVM) algorithm clearly performed better at predicting the accurate water content in the leaves for 4 days.

The work proposed in this paper will process the optical images acquired by UAV by data augmentation for the crop class with very few images. The processed images will be converted to grayscale downscaled to a low resolution. The textural features will be extracted from the grayscale images. Crop classification will be performed by using machine learning and deep learning algorithms for grayscale and textural-based images. With the evaluation measure, we will compare and evaluate the performance of how GLCM based textural features will outperform the ones with grayscale images. In this work, the main focus is how textural features will be helpful to distinguish between different types of crops compared to grayscale images. The paper is organized as follows, where a literature review is conducted in "Related Work", data set used in the study along with methodology is discussed in "Methodology", results and discussion in "Results and Discussion" and conclusion and future work in "Conclusion and Future Work".

## METHODOLOGY

Normalized Difference Vegetation Index (NDVI) is known as a standard index used in remote sensing for identifying chlorophyll content in an image based on Near Infra band (NIR) and Red (R) bands. It is quite challenging to differentiate the crop based on NDVI values, because various crops have similar profile. For instance, it is hard to discriminate between wheat and maize crop based on NDVI profile acquired from satellite imagery. In order to address the problem of differentiating different crops based on NDVI profile, UAV optical imagery is collected and GLCM features are extracted from the images. In this study, machine learning and deep learning algorithms are applied for classification of crops and GLCM-based texture features are used as an additional features to help in the classification. A comparison is also performed between classification with gray scale based images and GLCM features based classification. This section provide the details of study area for experiment and then discusses the methodology based on modules of our experiment.

### Study area & data set

To perform crop classification, an experimental area in the capital of Pakistan, Islamabad located at the National Agriculture Research Center (NARC) is selected. In the NARC region, various types of crops are grown throughout the year and experiments are performed. For our research, we selected four crops wheat, maize, rice, and soybean as shown in Fig. 1. The crop calendar for Pakistan can be viewed at *Crop Calendar (2020)*, where the particular locations of the crops in the study along with their growth cycle is enlisted in Table 1. The climate of Islamabad is a humid subtropical climate with an average rainfall of 790.8 mm.

The data set used in the study was gathered for five different crops at the different growth cycles of crops as shown in Table 1 using DJI Phantom pro-Advanced. All the selected crops including wheat, rice, soybean, and maize have overlapping crop cycles,

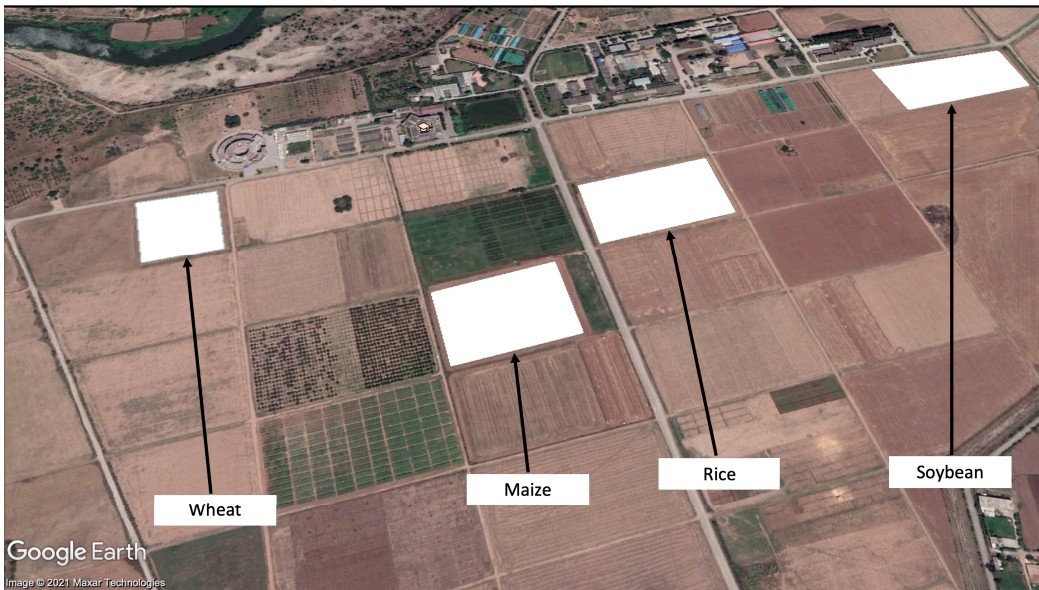

**Figure 1 Crops marked in © Google Earth (NARC Region).**

**Table 1 List of crops selected in study area.**

| Crop | Crop-cycle | Location |
| --- | --- | --- |
| Wheat-I | Dec-18 to Jun-19 | 30°40′ 22.25″ N, 73°07′ 18.28″ E |
| Rice | Jun-19 to Oct-19 | 30°40′ 25.19″ N, 73°07′ 27.93″ E |
| Soybean | Jul-19 to Dec-19 | 33°40′ 34.46″ N, 73°08′ 10.20″ E |
| Wheat-II | Nov-19 to May-20 | 33°40′ 17.29″ N, 73°07′ 48.98″ E |
| Maize | Mar-19 to Jul-19 | 33°40′ 18.69″ N, 73°07′ 37.84″ E |

especially winter wheat crop and winter maize crop had the same planting time. It was quite challenging to separate wheat and maize crops based solely on their NDVI profile. In order to address this problem, UAV optical imagery was collected and GLCM features were extracted from these images. Subsequently, several machine/deep learning models were applied to perform crop classification where the details of the dataset acquisition, machine learning/deep models, results are provided in the following sections.

Figure 2 shows the architectural diagram of the system divided into modules. The first module is the data acquisition where the data is collected with the help of a UAV drone. After the collection of the data, the next step is the pre-processing of the data, which requires analysis to remove images outside the boundary of the crop and to apply data augmentation for the crops fields where we have a limited amount of data. The next step is feature extraction where we will extract features from the grayscale images which can help in the study. The last step is to classify the crop classification based on machine learning and deep learning algorithms for gray scale-based images and feature-based images and to evaluate the result of crop classification. Each module is discussed in detail in this section.

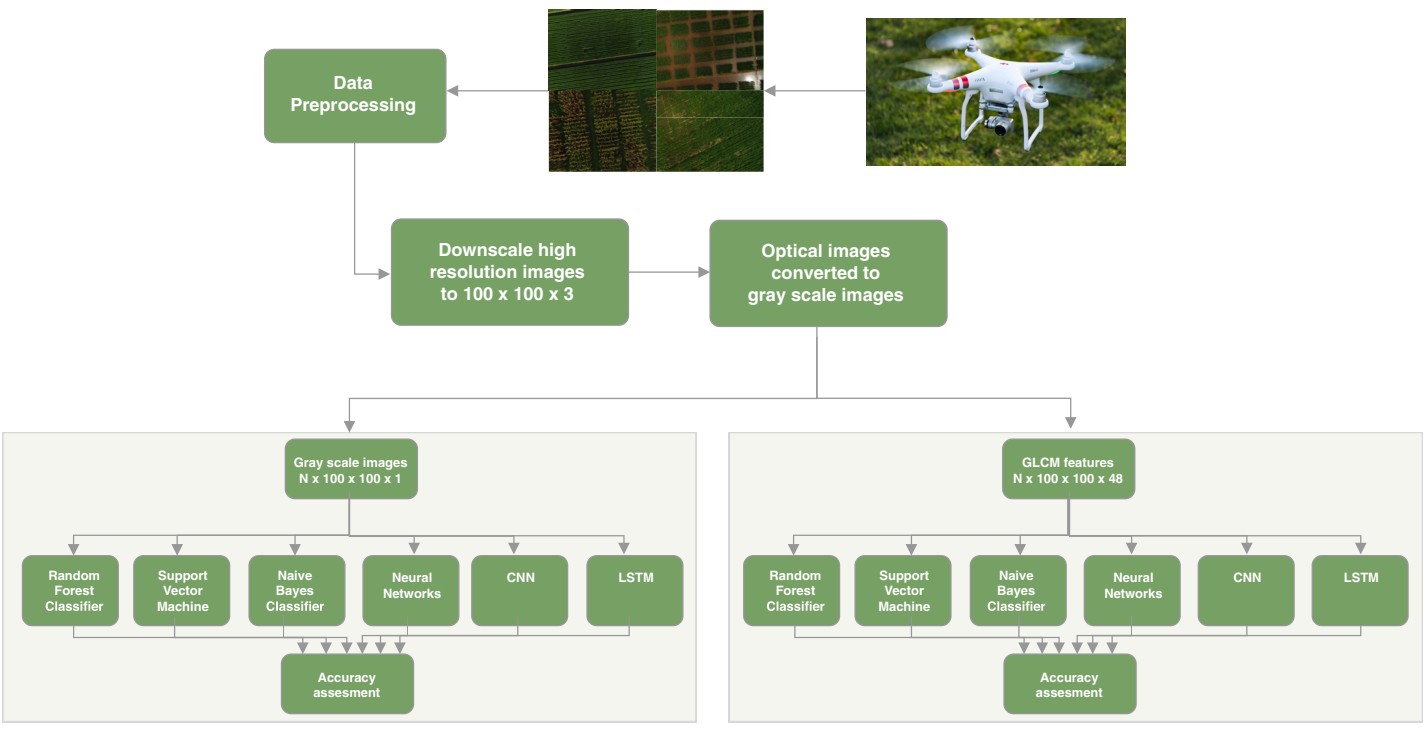

**Figure 2 System architecture.**

**Table 2 Specifications of UAV drone used in the study.**

| Characteristics | Technical specifications |
| --- | --- |
| Type | Four-rotor electric UAV |
| Weight | 1,368 g |
| Manufacturer | DJI |
| Model | FC6310 |
| Operating Temperature | 0° to 40° |
| Camera Sensor | 1″ CMOS |
| Image Size | 4,864 × 3,648 |
| Flight Duration | 30 min |
| Battery | 5,870 mAH LIPo 4S |

UAV platforms provide the ability to gather images at higher spatial resolution compared to satellite-based solutions. In this study, the DJI Phantom pro-Advanced (details mentioned in Table 2) equipped with a 20 Megapixel camera is used for data acquisition.

The data was collected by carrying out multiple flights to cover five fields at different stages of the crop cycle which are listed in Table 3. The first wheat flight was conducted on 16-May-2019 with the wheat field at max maturity stage and the second wheat crop flight was performed on 02-March-2020 at tillering stage. The flight for soybean was

| Table 3 Crop fields images acquired at various stage of crop cycle. | | | | | |
|---|---|---|---|---|---|
| Crop | Stage | Acquisition date | Acquisition time | Altitude | Images count |
| Wheat-I | Max Maturity | 16-May-2019 | 12:20 PM | 70 foot | 41 |
| Rice | Max-Tiller | 03-Sept-2019 | 12:15 PM | 120 foot | 3 |
| Soybean | V2 Stage | 03-Sept-2019 | 12:40 PM | 70 foot | 20 |
| Wheat-II | Tiller Stage | 02-March-2020 | 01:30 PM | 70 foot | 20 |
| Maize | Max Maturity | 24-July-2019 | 01:15 PM | 70 foot | 39 |

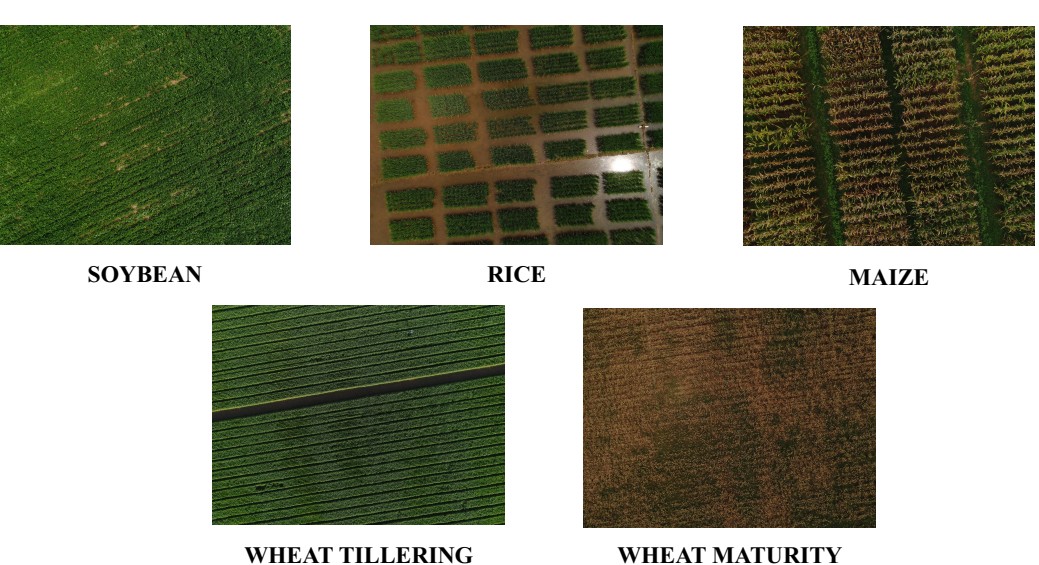

SOYBEAN          RICE          MAIZE

WHEAT TILLERING          WHEAT MATURITY

**Figure 3 Crops optical images captured by using DJI Phantom.**

conducted on 03-September-2019, whereas the flight for the rice field was done on 03-September-2019 at the max-tiller stage. The flight for the maize crop was done at the max-maturity stage on 24-July-2019. Due to limited images of the rice field, the images of the rice field were augmented to make the images count equivalent to the minimum of the rest of the crop field classes. Figure 3 shows the five crop fields including soybean, rice, maize, wheat at tillering stage and wheat at the maturity stage.

## Data pre-processing

The first step after collecting the data is to pre-process it to make it suitable for training. The captured images were analyzed without removing the images outside of the boundary. The collected data was organized in folders containing the date of the collection along with the stage of the particular crops. In order to perform supervised classification, a field survey was conducted to label each image with the help of NARC experts. Initially, the collected data was not sufficient to apply any classification technique, therefore, data augmentation was used to enhance the data. For this purpose, horizontal flipping and zoom with a minor factor was applied using Keras pre-processing library (*Data Augmentation, 2020*). The optical images captured with the drone were high-resolution

images, and performing classification requires computing power. To reduce the processing requirement, the images were downscaled to a size of $100 \times 100$, and the features were extracted from the down-scaled images.

## Feature extraction

The pixels in an optical image contained noise which was reduced by the information from texture and was considered as a complementary feature in classification. The analysis of textural images was categorized into four different groups that include model-based, statistical, geometrical, and signal processing (*Tuceryan & Jain, 1993*). For feature extraction, GLCM was employed, which is a widely used statistical technique developed by *Haralick, Shanmugam & Dinstein (1973)* for processing of remote sensing data. In the first step, the original image was converted to the grayscale. The next step was to extract spatial features from the gray-scale images based on the relationship of brightness values to the center pixel with its neighborhood defined by a kernel or window size. The relationship of the brightness values was represented in the form of a matrix. The matrix was made up of the frequent occurrence of the sequential pair of the pixel values along with a defined direction. The relationship helps GLCM to generate a different set of texture information based on gray-scale, kernel size, and direction. Harlick in *Haralick, Shanmugam & Dinstein (1973)* defined fourteen textural features, which provide redundant spatial context information which was an overhead in classification. In this study, only six textural features are considered which are listed below:

- Contrast (CON): The contrast measures the spatial frequency of an image and is a different moment of GLCM. It is the difference between the highest and the lowest values of the adjacent set of pixels. The contrast texture measures the local variations present in the image. An image with the low contrast presents GLCM concentration term around the principal diagonal and features low spatial frequencies.
- Homogeneity (HOM): This statistical measure is also called inverse difference moment. It measures the homogeneity in the image where it assumes larger values for smaller differences in grey tone within-pair elements. Homogeneity is more sensitive to the presence of near diagonal elements in the GLCM. The value of homogeneity is maximum when elements in the image are the same. GLCM contrast and homogeneity are strongly but inversely correlated, which means homogeneity decreases when contrast increases while energy is kept constant.
- Dissimilarity (DIS): Dissimilarity is a linear measure of local variations in an image.
- Angular second Moment (ASM): It measures textural uniformity i.e. repetitions in pixel pair. It detects the disorders in textures of the images. The maximum value achieved by the angular second moment is one. Higher values occur when the gray level distribution has a constant periodic form.
- Energy (EG): Energy is computed as the square root of an angular second moment. When the window is orderly, energy has higher values.
- Correlation (CORR): It is a measure of linear dependencies between the gray tone of the image.

Each of the listed textural feature is computed using the Eqs. (1) to (6) (*GLCM Equations, 2011*):

$$CON = \sum_{i=0}^{N-1} \sum_{j=0}^{N-1} (i-j)^2 \tag{1}$$

$$HOM = \sum_{i=0}^{N-1} \sum_{j=0}^{N-1} \frac{P(i,j)}{1+(i-j)^2} \tag{2}$$

$$DIS = \sum_{i=0}^{N-1} \sum_{j=0}^{N-1} P(i,j) x |i-j| \tag{3}$$

$$ASM = \sum_{i=0}^{N-1} \sum_{j=0}^{N-1} P(i,j)^2 \tag{4}$$

$$EG = \sqrt{\sum_{i=0}^{N-1} \sum_{j=0}^{N-1} P(i,j)^2} \tag{5}$$

$$CORR = \sum_{i=0}^{N-1} \sum_{j=0}^{N-1} \frac{(i-\mu_i)(j-\mu_j)}{\sqrt{(\sigma_i)(\sigma_j)}} \tag{6}$$

where N denotes the number of gray levels, while $P(i,j)$ is the normalized value of the gray-scale at position $i$ and $j$ of the kernel with a sum equal to 1. The textural features were generated from $100 \times 100$ gray-scale images. In this study, the kernel size was set to 19, and a total of 48 features were generated for each gray-scale image with distance at 1 and 2, rotation at 0, 45°, 90°, and 135° for each textural feature.

## Crop classification

In order to perform crop classification on the collected dataset, several supervised techniques are applied which are discussed below.

### Naive Bayes classifier

Naive Bayes Classifier is a simple probabilistic classifier that is based on the Bayes theorem. The inducer in Naive-Bayes computes conditional probabilities of classes given the instance and selects the class with higher posterior probability (*Witten & Frank, 2002*).

### Neural network

Neural Network is a very famous model which is designed to mimic the human brain to perform classification and regression tasks. It contains one input layer, or more hidden layers where each layer holds several neurons or nodes, and a single output layer (*Goodfellow et al., 2016*). Each layer computes some mathematical functions which enable it to find the complex relationship in the data.

### Support vector machines

The goal of the Support Vector Machine (SVM) is to find an optimal boundary that separates the classes based on data in the training set (*Ding, Qi & Tan, 2011*). SVM

algorithm solves the optimization in a way that it tends to maximize the margin between decision boundary (*Gunn, 1998*).

### Random forest classifier

Random Forest Classifier is developed by *Breiman (2001)* which performs classification by extending decision to multiple trees instead of a single tree. The ability to diversify through multiple trees instead of a single tree helps to achieve better classification performance. The final class of the particular instance is decided by the majority votes of all trees. Random Forest requires only a few parameters including the number of variables required for partitioning of the nodes and the number of trees to be grown.

### Convolutional Neural Network (CNN)

CNN is a deep learning model which is commonly used on imagery data (*Goodfellow et al., 2016*). It consists of an input layer, multiple hidden layers and an output layer where hidden layers are comprised of convolutional layers followed by the pooling layer and dense layer.

### Long Short Term Memory (LSTM) netwrok

LSTM is another deep learning model based on the Recurrent Neural Network (RNN) which has the capability to learn from the time series data with long dependencies (*Goodfellow et al., 2016*). Each layer in the LSTM model is the set of recurrently connected blocks or memory cells that performs reasonably well on several complex tasks such as crop classification.

## Evaluation metrics

The evaluation metrics used to assess the performance of the machine and deep learning algorithms are described as follows:

### Producer accuracy

Producer's Accuracy (PA) defined in Eq. (7) is the accuracy map from the point of reference of the producer. The *PA* shows how the classified map depicts the real features on the ground correctly or the probability that how certain land cover of an area on the ground is classified. The *PA* is the complement of the Omission Error (OE), where *PA* = 100% − *OE* (*Story & Congalton, 1986*).

$$PA = \frac{No.\ of\ correctly\ classified\ images\ of\ a\ class}{Total\ no.\ of\ ground\ truth\ images\ for\ class} \tag{7}$$

### User accuracy

The User's Accuracy (UA) defined in Eq. (8) is the accuracy with respect to the user of the map. The UA shows how often the class on the classification map will actually be in the ground data. The UA is the complement of the Commission Error (CE), *UA* = 100% − *CE*. *UA* is calculated by taking a total number of correct classifications for a class divided by the total number of the class.

$$UA = \frac{No.\ of\ correctly\ classified\ images\ of\ a\ class}{Total\ no.\ of\ images\ classified\ for\ class} \tag{8}$$

### Overall accuracy

Overall Accuracy (OAA) defined in Eq. (9) essentially tells us out of all available classes from classification what proportion are classified correctly. The overall accuracy is usually expressed in percentage, where 100% depicts all classes in the classification classified correctly.

$$OAA = \frac{Total\ no.\ correctly\ classified\ images\ of\ all\ classes}{Total\ no.\ of\ ground\ truth\ images\ of\ all\ classes} \tag{9}$$

### Precision

Precision refers to the number of positive class instances correctly classified out of total classified instances of the class. The formula to compute precision is defined in Eq. (10), where TP means true positive and FP means false positive.

$$Precision = \frac{TP}{TP + FP} \tag{10}$$

### Recall

Recall refers to amount of total instances of a positive class that are classified correctly. The formula to compute recall is defined in Eq. (11), where TP means true positive and FN means false negative.

$$Recall = \frac{TP}{TP + FN} \tag{11}$$

### F1-score

F1-Score is computed to provide a balance between precision and recall, compared to individually computing precision and recall does not cover all aspects of accuracy. F1-Score is calculated using the Eq. (12). The range of F1-score is between 0 and 1, where higher number shows the higher performance of the particular model.

$$F1 - Score = \frac{2 * Precision * Recall}{Precision + Recall} \tag{12}$$

### Accuracy

Accuracy refers to capability of the model to produce correct predictions for the instances observed. It is defined in Eq. (13), where TP means true positive, TN means true negative, FP means false positive and FN means false negative.

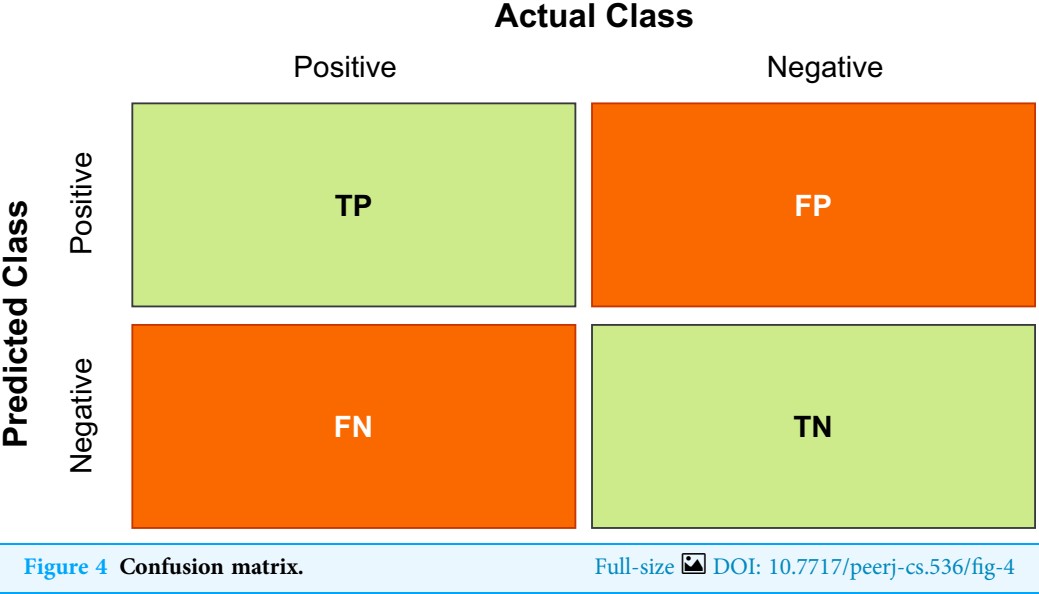

**Figure 4**  Confusion matrix.

$$Accuracy = \frac{TP + TN}{TP + TN + FP + FN} \tag{13}$$

All the TP, TN, FP and FN values can easily be computed by drawing the confusion matrix which is visual representation of all these values as shown in Fig. 4

Figure 4 shows the confusion matrix for two classes i.e. Positive and Negative. The TP is the correctly classified tuples of class Positives, TN is the number of tuples that are correctly classified as Negative. However, FP is the number of Negative tuples which are incorrectly classified as Positive. Similarly, FN is the number of Positive tuples that are wrongly classified ad Negative (*Kantardzic, 2011*). In the crop classification domain, the confusion matrix is another metric that is used to see the performance of the model in detail.

## RESULTS AND DISCUSSION

The machine learning and deep learning algorithms used in our study were support vector machine (SVM), random forest classifier (RFC), naive Bayes classifier, and neural networks (NN). Each algorithm was applied once on grayscale images and once on the images with GLCM based textural features. We have selected five crop classes at various phenological stages of the crops. The results of the overall accuracy of all classes and individual crop class performance are organized in separate tables to give a better overview of the performance. Table 4 shows confusion matrix for classification performed on the grayscale images using SVM algorithm. The SVM algorithm was able to classify rice and wheat-T crop correctly. The algorithm with only grayscale images was not able to classify soybean crop correctly and all the images of soybean were classified as wheat (maturity stage) crop. SVM algorithm correctly classifies 93% of wheat crop images and only 7% images were classified as maize, similarly, 73% of maize crop images were correctly

**Table 4 Confusion matrix for classification performed on grayscale images using SVM.**

| Class | Soybean | Rice | Wheat-T | Wheat | Maize | PA (%) |
|---|---|---|---|---|---|---|
| Soybean | 0 | 0 | 0 | 9 | 0 | 0 |
| Rice | 0 | 5 | 0 | 0 | 0 | 100 |
| Wheat-T | 0 | 0 | 5 | 0 | 0 | 100 |
| Wheat | 0 | 0 | 0 | 13 | 1 | 92.9 |
| Maize | 0 | 0 | 3 | 0 | 8 | 72.7 |
| UA (%) | 0 | 100 | 62.5 | 59.1 | 88.9 | |
| OAA (%) | 70.45% | | | | | |

**Table 5 Confusion matrix for classification performed on generated textures features images using SVM.**

| Class | Soybean | Rice | Wheat-T | Wheat | Maize | PA (%) |
|---|---|---|---|---|---|---|
| Soybean | 6 | 0 | 0 | 3 | 0 | 66.7 |
| Rice | 0 | 5 | 0 | 0 | 0 | 100 |
| Wheat-T | 0 | 0 | 5 | 0 | 0 | 100 |
| Wheat | 1 | 1 | 0 | 12 | 0 | 85.7 |
| Maize | 0 | 1 | 1 | 0 | 9 | 81.8 |
| UA (%) | 85.7 | 71.4 | 83.3 | 80 | 100 | |
| OAA (%) | 84.1% | | | | | |

classified and only 27% images were classified as wheat-T. The overall accuracy obtained by SVM classification on grayscale images was 70.45%.

Table 5 shows the confusion matrix for classification performed on generated GLCM textural features using SVM algorithm. The SVM algorithm on GLCM based textural features was able to classify rice and wheat-T crop images correctly. The algorithm with GLCM based textural images was able to classify 66.67% soybean crop images correctly, whereas 33.33% images were classified as wheat (maturity stage). Similarly, the algorithm classifies 85.7% wheat crop images correctly, whereas 7.1% images were classified as rice and the remaining 7.1% images were classified as soybean. The algorithm classifies 81.82% maize crop images correctly, whereas 9.1% images were classified as rice and the remaining 9.1% images were classified as wheat-T. The overall accuracy obtained for SVM classification on generated GLCM based textural features was 84.10% which showed an improvement of 13.65% compared to training the classifier only on grayscale images. This improvement clearly shows the impact of using textural features extracted from grayscale images and their ability to distinguish between different crop types.

Table 6 shows the confusion matrix for classification performed on grayscale images using Random Forest classifier. The Random Forest classifier was able to classify rice, wheat, wheat-T, and maize crop correctly. The algorithm with only grayscale images was not able to classify soybean crop correctly, except one soybean image all the remaining

**Table 6 Confusion matrix for classification performed on grayscale images using Random Forest Classifier.**

| Class | Soybean | Rice | Wheat-T | Wheat | Maize | PA (%) |
|---|---|---|---|---|---|---|
| Soybean | 1 | 0 | 0 | 8 | 0 | 12.5 |
| Rice | 0 | 5 | 0 | 0 | 0 | 100 |
| Wheat-T | 0 | 0 | 5 | 0 | 0 | 100 |
| Wheat | 0 | 0 | 0 | 14 | 0 | 100 |
| Maize | 0 | 0 | 0 | 0 | 11 | 100 |
| UA (%) | 100 | 100 | 100 | 63.6 | 100 | |
| OAA (%) | 81.82% | | | | | |

**Table 7 Confusion matrix for classification performed on GLCM features using Random Forest Classifier.**

| Class | Soybean | Rice | Wheat-T | Wheat | Maize | PA (%) |
|---|---|---|---|---|---|---|
| Soybean | 5 | 0 | 0 | 4 | 0 | 55.6 |
| Rice | 0 | 5 | 0 | 0 | 0 | 100 |
| Wheat-T | 0 | 0 | 5 | 0 | 0 | 100 |
| Wheat | 0 | 0 | 0 | 14 | 0 | 100 |
| Maize | 0 | 0 | 0 | 0 | 11 | 100 |
| UA (%) | 100 | 100 | 100 | 77.8 | 100 | |
| OAA (%) | 90.91% | | | | | |

images of soybean were classified as wheat (maturity stage) crop. The overall accuracy obtained by the Random Forest classifier on the grayscale images was 81.82%.

Table 7 shows the confusion matrix for the classification performed on generated GLCM textural features using Random Forest classifier. The Random Forest classifier based on GLCM based textural features was able to classify rice, wheat, wheat-T, and maize crop correctly. The algorithm with GLCM based textural images was able to classify 55.56% soybean crop images correctly, where 44.4% images were classified to wheat (maturity stage) images. The overall accuracy obtained for the Random Forest classifier on generated GLCM based textural features was 90.91% which showed an improvement of 9.09% compared to training the classifier only on grayscale images. This improvement clearly indicates the ability of the textural features extracted from grayscale images to distinguish between the crop types.

Table 8 shows the confusion matrix for classification performed on grayscale images using Naive Bayes classifier. The Naive Bayes classifier was able to classify rice, wheat (maturity stage), wheat (tillering stage), and maize crop correctly. The classifier with only grayscale images was not able to classify soybean crop correctly and all the images of soybean were classified as wheat (maturity stage) crop. The overall accuracy obtained for Random Forest classification on grayscale images was 79.55%. Table 9 shows the confusion matrix for classification performed on generated GLCM textural features using Naive Bayes classifier. The Naive Bayes classifier based on GLCM based textural features was able

**Table 8 Confusion matrix for classification performed on gray scale images using Naive Bayes Classifier.**

| Class | Soybean | Rice | Wheat-T | Wheat | Maize | PA (%) |
|---|---|---|---|---|---|---|
| Soybean | 0 | 1 | 1 | 7 | 0 | 0 |
| Rice | 0 | 5 | 0 | 0 | 0 | 100 |
| Wheat-T | 0 | 0 | 5 | 0 | 0 | 100 |
| Wheat | 0 | 0 | 0 | 14 | 0 | 100 |
| Maize | 0 | 0 | 0 | 0 | 11 | 100 |
| UA (%) | 0 | 83.3 | 83.3 | 66.7 | 100 | |
| OAA (%) | 79.55% | | | | | |

**Table 9 Confusion matrix for classification performed on generated textures features images using Naive Bayes Classifier.**

| Class | Soybean | Rice | Wheat-T | Wheat | Maize | PA (%) |
|---|---|---|---|---|---|---|
| Soybean | 5 | 1 | 0 | 3 | 0 | 55.6 |
| Rice | 0 | 5 | 0 | 0 | 0 | 100 |
| Wheat-T | 0 | 0 | 5 | 0 | 0 | 100 |
| Wheat | 0 | 0 | 0 | 14 | 0 | 100 |
| Maize | 0 | 0 | 0 | 0 | 11 | 100 |
| UA (%) | 100 | 83.3 | 100 | 82.4 | 100 | |
| OAA (%) | 90.91% | | | | | |

to classify rice, wheat (maturity stage), wheat (tiler stage), and maize crop correctly. The algorithm classified 55.56% soybean crop images correctly, where 11.11% images were classified as rice crop and 33.33% images were classified as wheat (maturity stage) images. The overall accuracy obtained for the Naive Bayes classifier on generated GLCM based textural features was 90.91% which showed an improvement of 11.36% compared to training the classifier on grayscale images only. This improvement clearly shows the impact of textural features extracted from grayscale images for distinguishing the crop types. Table 10 shows the confusion matrix for classification performed on grayscale images using a feed-forward neural network classifier. The feed-forward neural network classifier was able to classify wheat crop correctly. The classifier with only grayscale images classifies soybean, wheat-T, wheat, maize as the wheat crop. Also, the rice crop images were classified as maize crop. The overall accuracy obtained for feed-forward neural network classifier on grayscale images was 31.82%. The reason for this poor performance was the limited amount of available data which was not enough to train a deep learning model and thus resulted in showing an average performance. Table 11 shows the confusion matrix for classification performed on generated GLCM textural features using feed-forward neural network classifier. The feed-forward neural network classifier was able to classify maize crop only. The classifier with only GLCM based images classifies all crop images into a single class. Each of the soybean, rice, wheat-T, wheat, and maize images were classified as a maize crop. The overall accuracy obtained for feed-

Table 10 Confusion matrix for classification performed on gray scale images using Neural Networks.

| Class | Soybean | Rice | Wheat-T | Wheat | Maize | PA (%) |
|---|---|---|---|---|---|---|
| Soybean | 0 | 0 | 0 | 9 | 0 | 0 |
| Rice | 0 | 0 | 0 | 0 | 5 | 0 |
| Wheat-T | 0 | 0 | 0 | 5 | 0 | 0 |
| Wheat | 0 | 0 | 0 | 14 | 0 | 100 |
| Maize | 0 | 0 | 0 | 11 | 0 | 0 |
| UA (%) | 0 | 0 | 0 | 35.9 | 0 | |
| OAA (%) | 31.82% | | | | | |

Table 11 Confusion matrix for classification performed on generated textures features images using Neural Networks based classifier.

| Class | Soybean | Rice | Wheat-T | Wheat | Maize | PA (%) |
|---|---|---|---|---|---|---|
| Soybean | 0 | 0 | 0 | 0 | 9 | 0 |
| Rice | 0 | 0 | 0 | 0 | 5 | 0 |
| Wheat-T | 0 | 0 | 0 | 0 | 5 | 0 |
| Wheat | 0 | 0 | 0 | 0 | 14 | 0 |
| Maize | 0 | 0 | 0 | 0 | 11 | 100 |
| UA (%) | 0 | 0 | 0 | 0 | 25 | |
| OAA (%) | 25% | | | | | |

Table 12 Confusion matrix for classification performed on gray scale images using LSTM.

| Class | Soybean | Rice | Wheat-T | Wheat | Maize | PA (%) |
|---|---|---|---|---|---|---|
| Soybean | 0 | 0 | 0 | 9 | 0 | 0 |
| Rice | 0 | 5 | 0 | 0 | 0 | 100 |
| Wheat-T | 0 | 1 | 0 | 4 | 0 | 0 |
| Wheat | 0 | 0 | 0 | 13 | 1 | 93 |
| Maize | 0 | 0 | 0 | 0 | 11 | 100 |
| UA (%) | 0 | 83 | 0 | 50 | 92 | |
| OAA (%) | 65.91% | | | | | |

forward neural network classifier on GLCM based images was 25% and was lower compared to the grayscale images. The reason for this poor performance was the limited amount of available data which was not enough to train a deep learning model and thus resulted in showing an average performance.

Table 12 shows the confusion matrix for classification performed on grayscale images using long short term memory (LSTM). The LSTM based classifier when applied on the grayscale image was able to classify maize and rice crop correctly. The classifier with only grayscale images classifies soybean and wheat-T as a wheat crop at a mature stage. Also, the wheat crop images at the maturity stage were classified as wheat crop correctly,

**Table 13 Confusion matrix for classification performed on generated textures features images using LSTM.**

| Class | Soybean | Rice | Wheat-T | Wheat | Maize | PA (%) |
|---|---|---|---|---|---|---|
| Soybean | 0 | 0 | 0 | 0 | 9 | 0 |
| Rice | 0 | 0 | 0 | 0 | 5 | 0 |
| Wheat-T | 0 | 0 | 0 | 0 | 5 | 0 |
| Wheat | 0 | 0 | 0 | 0 | 14 | 0 |
| Maize | 0 | 0 | 0 | 0 | 11 | 100 |
| UA (%) | 0 | 0 | 0 | 0 | 25 | |
| OAA (%) | 25% | | | | | |

**Table 14 Confusion matrix for classification performed on gray scale images using CNN.**

| Class | Soybean | Rice | Wheat-T | Wheat | Maize | PA (%) |
|---|---|---|---|---|---|---|
| Soybean | 0 | 0 | 0 | 9 | 0 | 0 |
| Rice | 0 | 5 | 0 | 0 | 0 | 100 |
| Wheat-T | 0 | 0 | 5 | 0 | 0 | 100 |
| Wheat | 0 | 0 | 0 | 13 | 1 | 93 |
| Maize | 0 | 0 | 0 | 0 | 11 | 100 |
| UA (%) | 0 | 100 | 100 | 59 | 92 | |
| OAA (%) | 77.27% | | | | | |

but only one image was misclassified as maize crop. The overall accuracy obtained for LSTM based classifier on grayscale images was 65.91%.

Table 13 shows the confusion matrix for classification performed on generated GLCM textural features using long short term memory classifier. The LSTM based classifier was able to classify maize crop only. The classifier with only GLCM based images classifies all crop images into a single class. Each of the soybean, rice, wheat-T, wheat, and maize images were classified as maize crop. The overall accuracy obtained for LSTM based classifier on GLCM based images was 25% and was lower compared to the grayscale images. The reason for this poor performance was the limited amount of available data which was not enough to train a deep learning model and thus resulted in showing an average performance. Table 14 shows the confusion matrix for classification performed on grayscale images using convolutional neural network (CNN). The CNN-based classifier when applied on a gray scale image was able to classify rice, wheat-tiller, and maize crop correctly. The classifier with only grayscale images classifies soybean as a wheat crop. Also, the wheat crop images at the maturity stage were classified as wheat crop correctly, but only one image was misclassified as maize crop. The overall accuracy obtained from a CNN-based classifier on grayscale images was 77.27%.

Table 15 shows the confusion matrix for classification performed on generated GLCM textural features using convolutional neural network (CNN) based classifier. The CNN-based classifier was able to classify maize crop only. The classifier with GLCM based

**Table 15 Confusion matrix for classification performed on generated textures features images using CNN.**

| Class | Soybean | Rice | Wheat-T | Wheat | Maize | PA (%) |
|---|---|---|---|---|---|---|
| Soybean | 0 | 0 | 0 | 0 | 9 | 0 |
| Rice | 0 | 0 | 0 | 0 | 5 | 0 |
| Wheat-T | 0 | 0 | 0 | 0 | 5 | 0 |
| Wheat | 0 | 0 | 0 | 0 | 14 | 0 |
| Maize | 0 | 0 | 0 | 0 | 11 | 100 |
| UA (%) | 0 | 0 | 0 | 0 | 25 | |
| OAA (%) | 25% | | | | | |

**Table 16 Precision, Recall & F1-Score on gray scale images and texture images using SVM.**

| Class | Accuracy (%) | | Precision | | Recall | | F-1 Score | |
|---|---|---|---|---|---|---|---|---|
| | Gray scale | GLCM | Gray scale | GLCM | Gray scale | GLCM | Gray scale | GLCM |
| Soybean | 79.55 | 91.11 | 0.0 | 0.67 | 0.0 | 0.86 | 0.0 | 0.75 |
| Rice | 100 | 95.56 | 1.0 | 1.0 | 1.0 | 0.71 | 1.0 | 0.83 |
| Wheat-T | 93.18 | 97.78 | 1.0 | 1.0 | 0.63 | 0.83 | 0.77 | 0.91 |
| Wheat | 77.27 | 86.67 | 0.93 | 0.80 | 0.59 | 0.80 | 0.72 | 0.80 |
| Maize | 90.91 | 93.33 | 0.73 | 0.82 | 0.89 | 0.90 | 0.80 | 0.86 |

images classifies all crop images into a single class. Each of the soybean, rice, wheat-T, wheat, and maize images were classified as a maize crop. The overall accuracy obtained for CNN-based classifier on GLCM based images was 25% and was lower compared to the grayscale images. The reason for this poor performance was the limited amount of available data which was not enough to train a deep learning model and thus resulted in showing an average performance. The CNN-based classifier on GLCM generated images failed to learn any information.

It can be concluded from the results obtained by applying machine and deep learning algorithms that machine learning models with the help of textural features extraction using GLCM are able to outperform deep learning algorithms because of the limited data set available. To further enhance the performance of deep learning algorithms, there is a need to gather more data in order to achieve better results compared to machine learning algorithms.

Table 16 describes the accuracy, precision, recall and F1-score for SVM on the greyscale images and the texture images. The highest accuracy (100%) was achieved for rice crops using grayscale images. Similarly, F1-score was the highest for rice crops when SVM was applied. The highest value of precision was achieved for rice and wheat-T crops using grayscale images and texture-based images. The recall shows the highest value for rice using grayscale images.

Similarly, Table 17 shows the accuracy, precision, recall, and F1-score for the grayscale images and the texture images when Random forest was applied. The highest accuracy that

**Table 17 Precision, Recall & F1-Score on gray scale images and texture images using Random Forest Classifier.**

| Class | Accuracy (%) | | Precision | | Recall | | F-1 Score | |
|---|---|---|---|---|---|---|---|---|
| | Gray scale | GLCM | Gray scale | GLCM | Gray scale | GLCM | Gray scale | GLCM |
| Soybean | 81.82 | 90.91 | 0.11 | 0.56 | 1.0 | 1.0 | 0.20 | 0.71 |
| Rice | 100 | 100 | 1.0 | 1.0 | 1.0 | 1.0 | 1.0 | 1.0 |
| Wheat-T | 100 | 100 | 1.0 | 1.0 | 1.0 | 1.0 | 1.0 | 1.0 |
| Wheat | 81.82 | 90.91 | 1.0 | 1.0 | 0.64 | 0.78 | 0.78 | 0.88 |
| Maize | 100 | 100 | 1.0 | 1.0 | 1.0 | 1.0 | 1.0 | 1.0 |

**Table 18 Precision, Recall & F1-Score on gray scale images and texture images using Naive Bayes Classifier.**

| Class | Accuracy (%) | | Precision | | Recall | | F-1 Score | |
|---|---|---|---|---|---|---|---|---|
| | Gray scale | GLCM | Gray scale | GLCM | Gray scale | GLCM | Gray scale | GLCM |
| Soybean | 79.55 | 90.91 | 0.0 | 0.56 | 0.0 | 1.0 | 0.0 | 0.71 |
| Rice | 100 | 97.73 | 1.0 | 1.0 | 1.0 | 0.83 | 1.0 | 0.91 |
| Wheat-T | 100 | 100 | 1.0 | 1.0 | 1.0 | 1.0 | 1.0 | 1.0 |
| Wheat | 79.55 | 93.18 | 1.0 | 1.0 | 0.61 | 0.82 | 0.76 | 0.90 |
| Maize | 100 | 100 | 1.0 | 1.0 | 1.0 | 1.0 | 1.0 | 1.0 |

was 100% accuracy was obtained in the case of rice, maize, and wheat-T crops for grayscale and texture-based images. The F1-score for maize, rice, and wheat-T was highest in the case of both grayscale and texture images. The highest value of precision was observed for all crops with grayscale and texture-based images, whereas the highest precision was for the soybean crop.

Table 18 shows the accuracy, precision, recall, and F1-score for the grayscale images and the texture images when the Naive Bayes classifier was applied. The highest accuracy obtained for grayscale images was in the case of maize, wheat-T, and rice crop, whereas in the case of texture-based images wheat-T and rice crop accuracy were at the higher end. The F1-score for wheat-T and rice was the highest in the case of both grayscale and texture-based images. The highest value of precision was observed for all crops when textured-based images were used, except for soybean. Similarly, the Table 19 the performance summary of all evaluation metrics when Neural Network was applied on grayscale images and textural based images. The highest accuracy (100%) was observed for wheat-T and rice crop. The highest precision, F1-score, and recall were observed for wheat crop in case of grayscale images and maize crop in case of texture-based images. Table 20 shows the accuracy, precision, recall, and F1-score for the grayscale images and the texture images when LSTM based classifier was applied. The highest accuracy obtained for grayscale images was in the case of maize, wheat, wheat-T, and soybean crop, whereas in the case of texture-based images rice crop image accuracy was at the higher end. The F1-score for rice, wheat, and maize crop was higher for grayscale images. The highest

**Table 19 Precision, Recall & F1-Score on gray scale images and texture images using Neural Networks.**

| Class | Accuracy (%) | | Precision | | Recall | | F-1 Score | |
|---|---|---|---|---|---|---|---|---|
| | Gray scale | GLCM | Gray scale | GLCM | Gray scale | GLCM | Gray scale | GLCM |
| Soybean | 79.55 | 79.55 | 0.0 | 0.0 | 0.0 | 0.0 | 0.0 | 0.0 |
| Rice | 88.64 | 88.64 | 0.0 | 0.0 | 0.0 | 0.0 | 0.0 | 0.0 |
| Wheat-T | 88.64 | 88.64 | 0.0 | 0.0 | 0.0 | 0.0 | 0.0 | 0.0 |
| Wheat | 43.18 | 68.18 | 1.0 | 0.0 | 0.36 | 0.0 | 0.53 | 0.0 |
| Maize | 63.64 | 25 | 0.0 | 1.0 | 0.0 | 0.25 | 0.0 | 0.40 |

**Table 20 Precision, Recall & F1-Score on gray scale images and texture images using LSTM.**

| Class | Accuracy (%) | | Precision | | Recall | | F-1 Score | |
|---|---|---|---|---|---|---|---|---|
| | Gray scale | GLCM | Gray scale | GLCM | Gray scale | GLCM | Gray scale | GLCM |
| Soybean | 81.63 | 79.55 | 0.0 | 0.0 | 0.0 | 0.0 | 0.0 | 0.0 |
| Rice | 87.76 | 88.64 | 0.5 | 0.0 | 0.83 | 0.0 | 0.63 | 0.0 |
| Wheat-T | 89.8 | 88.64 | 0.0 | 0.0 | 0.0 | 0.0 | 0.0 | 0.0 |
| Wheat | 71.43 | 68.18 | 0.93 | 0.0 | 0.50 | 0.0 | 0.65 | 0.0 |
| Maize | 87.76 | 25 | 1.0 | 1.0 | 0.65 | 0.25 | 0.79 | 0.40 |

**Table 21 Precision, Recall & F1-Score on gray scale images and texture images using CNN.**

| Class | Accuracy (%) | | Precision | | Recall | | F-1 Score | |
|---|---|---|---|---|---|---|---|---|
| | Gray scale | GLCM | Gray scale | GLCM | Gray scale | GLCM | Gray scale | GLCM |
| Soybean | 81.63 | 79.55 | 0.0 | 0.0 | 0.0 | 0.0 | 0.0 | 0.0 |
| Rice | 89.8 | 88.64 | 0.5 | 0.0 | 1.0 | 0.0 | 0.67 | 0.0 |
| Wheat-T | 100 | 88.64 | 1.0 | 0.0 | 1.0 | 0.0 | 1.0 | 0.0 |
| Wheat | 79.59 | 68.18 | 0.93 | 0.0 | 0.59 | 0.0 | 0.72 | 0.0 |
| Maize | 87.76 | 25 | 1.0 | 1.0 | 0.65 | 0.25 | 0.79 | 0.40 |

value of precision and recall was observed for rice, wheat, and maize crops when gray scale-based images were used. Table 21 shows the accuracy, precision, recall, and F1-score for the grayscale images and the texture images when CNN based classifier was applied. The highest accuracy obtained for grayscale images was in the case of all the crops in classification i.e. soybean, rice, wheat, wheat-T, and maize crops. The F1-score for rice, wheat, wheat-T, and maize crop was higher for grayscale images. The highest value of precision and recall was observed for rice, wheat, wheat-T and maize crops when gray scale-based images were used.

In this particular analysis, crop classification was performed on the dataset collected by drone mounted with the optical camera. In order to perform crop classification, several machine learning algorithms i.e. Naive Bayes, random forest and support vector machine, feed-forward neural network; and deep learning algorithms i.e. Convolutional neural

network and LSTM were applied on grayscale images and GLCM based features extracted from these images. The result shows that the extracted GLCM features provide additional information which helps the machine/deep learning algorithms to learn better patterns and outperforms the simple gray scale-based classification. Among all other GLCM features, four GLCM features had a major impact on the classification results i.e. contrast, energy, dissimilarity, and angular second moment. Each of these four GLCM features tends to extract local variations in the image which helps the machine learning algorithms to perform better as compared to the normal gray scale-based classification.

Out of the four machine learning based algorithms applied on GLCM based images, random forest, and naive classifier achieved an overall accuracy of 90.9% whereas SVM-based classifier achieved only 84.1 which was still better compared to only using grayscale images for classification. The random forest algorithm performs better because each tree to be used, gives a single prediction for each record and it takes an average overall to give the final prediction which results in an overall better result. However, the deep learning models including CNN, LSTM don't perform well on the extracted GLCM based features. These models were designed to learn the features automatically using filters in their architecture as compared to the hand-engineered GLCM features. Moreover, the accuracy of these models doesn't improve on the grayscale images because of the very small dataset with a little variation which results in the miss classifies all crop images. The same was the case with the feed-forward neural network which shows poor performance with the accuracy of 63.64% on grayscale images and 25% on extracted GLCM features.

## Conclusion and Future Work

In this study, we investigated the potential of GLCM-based texture information for crop classification. The main goal of this study is to evaluate the benefit of textural features by comparing them with the grayscale images. The experimental area with five crops at different stages of the crop cycle is selected from the NARC fields located in Islamabad, Pakistan. The grayscale images have little information which makes them difficult to distinguish between the different classes. In contrast to this, the textural features extracted from the grayscale images show great potential to classify the different crop classes. Among these GLCM features, four features are found to be more useful for the particular experiment of crop classification including contrast, energy, dissimilarity, and angular second moment which try to extract local variations in the image.

In order to perform crop classification, machine/deep learning algorithms are applied including Naive Bayes, random forest, support vector machine, feed-forward neural network, convolutional neural network, and LSTM. The overall crop classification among the five classes, Naive Bayes, and random forest classifier for textural-based images achieved an accuracy of 90.91%. However, in the gray scale-based images, Naive Bayes achieved an accuracy of 79.55% and random forest achieved an accuracy of 81.82%. In contrast to this, the deep learning models including CNN, LSTM don't perform well on the extracted GLCM based features due to the small dataset. Similarly, the neural network achieved an accuracy of 31.82% in grayscale images and 25% in the case of textural-based

images. The deep learning algorithms will show a better performance for crop classification by using more data and with extracted texture-based images.

The major limitation of data acquisition is the specific altitude of the drone flight, which constrained us to cover all fields in one view. For this particular experiment, limited plots are chosen where images are collected from these allotted experimental fields. The other limitation is related to the drone flight i.e. drone cannot be flown at much higher altitude in the subject area due to security reasons.

To better analyze the results of crops grown in a region, the drone should be flown at an altitude of 400 ft covering at least 15 crop fields. In addition, for practical agriculture production, crop surface feature discrimination can be explored.

## ACKNOWLEDGEMENTS

The research work has been conducted in SEECS, NUST, in collaboration with NARC, Islamabad.

### Funding

The research work is funded by HEC under NRPU program, Pakistan. The funders had no role in study design, data collection and analysis, decision to publish, or preparation of the manuscript.

### Grant Disclosures

The following grant information was disclosed by the authors:
NRPU program, Pakistan.

### Competing Interests

The authors declare that they have no competing interests.

### Author Contributions

- Naveed Iqbal conceived and designed the experiments, performed the experiments, analyzed the data, performed the computation work, prepared figures and/or tables, authored or reviewed drafts of the paper, and approved the final draft.
- Rafia Mumtaz conceived and designed the experiments, performed the experiments, analyzed the data, authored or reviewed drafts of the paper, supervision, and approved the final draft.
- Uferah Shafi performed the experiments, analyzed the data, authored or reviewed drafts of the paper, and approved the final draft.
- Syed Mohammad Hassan Zaidi performed the experiments, analyzed the data, authored or reviewed drafts of the paper, and approved the final draft.

## Data Availability

The source code and the dataset used in the study is available at GitHub: https://github.com/niqbalmsit17/CropClassification.

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
