# Peer review of "Gray level co-occurrence matrix (GLCM) texture based crop classification using low altitude remote sensing platforms"

_PeerJ Computer Science, doi:10.7717/peerj-cs.536_

## Round 0.1 · original submission · Major Revisions

The topic is interesting but the reviewers have raised serious concerns on the experimental design, and validity of findings (particularly see comments of reviewers 2 and 3). Here are my further comments:

- Was a proper design of experiment technique followed? Which other methodology options were considered? Provide a justification for the selected methodology choice.

- The system architecture also needs to be explained further.

- The descriptions of evaluation metrics in section 5 may be moved to section 4.

- Findings/results (section 5) does not provide in depth discussion and analysis. Adding a new 'discussion' section would help, where results and research objectives are discussed in light of the literature, as well as limitations and future work are presented in detail.

- Tables 4-15 (section 5). It would be good to add how to read the confusion matrices.

Reviewer 1 ·

Basic reporting

the paper needs a grammar polishing. Please have it proofread carefully again.

Experimental design

The experiment needs to be more comprehensive, I would suggest to add deep learning algorithms such as CNN and LSTM. In addition, the paper should include comparision with state-of-the-art approaches

Validity of the findings

The dataset collection is very interesting and it can be helpful for future research direction

Additional comments

Gray level co-occurrence matrix (GLCM) based features are extracted from underlying gray scale images collected by the drone. To classify the different types of crops, different ML algorithms including Random Forest (RF), Naive Bayes (NB), Neural Network (NN) and Support Vector Machine (SVM) are applied.
How is the data annotation? Is the data labelled by someone who expert in agriculture?

I would recommend to use deep learning algorithms in this paper you can find information in the following paper:
Dashtipour, K., Gogate, M., Li, J., Jiang, F., Kong, B. and Hussain, A., 2020. A hybrid Persian sentiment analysis framework: Integrating dependency grammar based rules and deep neural networks. Neurocomputing, 380, pp.1-10.
I would recommend to use graph neural network:
Scarselli, F., Gori, M., Tsoi, A.C., Hagenbuchner, M. and Monfardini, G., 2008. The graph neural network model. IEEE Transactions on Neural Networks, 20(1), pp.61-80.
I would suggest to compare your results with state-of-the-art approaches such as:
Zahid, A., Abbas, H.T., Ren, A., Zoha, A., Heidari, H., Shah, S.A., Imran, M.A., Alomainy, A. and Abbasi, Q.H., 2019. Machine learning driven non-invasive approach of water content estimation in living plant leaves using terahertz waves. Plant Methods, 15(1), p.138.
Sun, J., Abdulghani, A.M., Imran, M.A. and Abbasi, Q.H., 2020, April. IoT Enabled Smart Fertilization and Irrigation Aid for Agricultural Purposes. In Proceedings of the 2020 International Conference on Computing, Networks and Internet of Things (pp. 71-75).

Reviewer 2 ·

Basic reporting

In this paper, the optical images were used to classify different crops using UAV. The topic was interesting. The used calibration methods were normally used methods, the experiment design was not quite practical in real applications.

Experimental design

From Table 1 and Figure 1, the classified crops (wheat, rice, soybean, maiz) were not planted at the same time. So the results of this classifiation was not a practical production scenario. We could classify the crops based on the planting time.
The detals of experiment should be supplied, such as planting time or imaging acquisition time, the flight information of UAV, the calibration board of UAV imaging and so on.

Validity of the findings

I think the methods used for crop classification is helpful, eventhough these methods were normally used methods.The authors just used for comparision, not any improvement for related methods.

Additional comments

I suggest to make a crop or surface features discrimination at the same time in practical agricultual productions, not just for the purpose of classification.

Reviewer 3 ·

Basic reporting

The conclusion can't represent the whole results. What reviewer can see is that the GLCM based classification outperform the grayscale based in some cases. And it's hard to get how the 13.5% is reached.

Experimental design

The experiment was conducted in seperated fields at different time. It can be more conclusive and usable if this classifier be used in a large orthoimage which contains different types of ground cover.

Validity of the findings

More detailed discussion and conclusion are needed since the result is realy ambiguous.

Additional comments

The topic of this paper is interesting but needs improvement. The conclusion in abstract can't find proof in results section.

---

## Round 0.2 · Minor Revisions

Thank you for revising the article in response to the comments of the reviewers and the editors.

1) Just to elaborate on my previous comment related to the design of experiments (DOE), please see following sources:
- Engineering Statistics Handbook, NIST, https://www.itl.nist.gov/div898/handbook/index.htm
https://www.itl.nist.gov/div898/handbook/pmd/section3/pmd31.htm
https://www.itl.nist.gov/div898/handbook/pri/section1/pri11.htm
- Dan Nettleton, A Discussion of Statistical Methods for Design and Analysis of Microarray Experiments for Plant Scientists, The Plant Cell Sep 2006, 18 (9) 2112-2121; https://doi.org/10.1105/tpc.106.041616.
- iSixSigma, https://www.isixsigma.com/tools-templates/design-of-experiments-doe/design-experiments-%E2%90%93-primer/
- ASQ, https://asq.org/quality-resources/design-of-experiments

It may be fine if you don't use the DOE technique for this research. However, it is important to provide the ratinale of choosing a certain methodology, and what choices were available. These are still not clearly presented in the methodology section, which directly starts by discussing the methodology steps.

Data set (section 3) should also be part of the methodology (section 4). The above questions also apply on the selection of the data set.

2) The article needs substantial English language improvements based on a complete check. Below are only some examples where such improvements are needed:

L474 the hand engineered GLCM features. Moreover, the accuracy of these models doesn’t improved on the
(doesn't improved --> don't improve; OR didn't improve)

L499 are acquired for the allotted field in the study. The second limitation the drone. In order to cover various
(The second limitation the drone. --> The second limitation was related to the drone.)

L333 The machine learning and deep learning algorithms used in our study are support vector machine (SVM)
(SVM has been used multiple times before L333, e.g. L50 (full form), L115, L165, L171, L183 (full form), L295 (full form), L296) - Please ensure that all abbreviations are used in full when these are first introduced, followed by use of their abbreviated forms.

Please also check the use of present vs past tenses throughout the article, e.g. is vs was, are vs were. This is very important as the wrong use would change the meaning at all, particularly when discussing the experiments. For example, the use of past tense may imply that some thing was done that way particulalry in your experiment while the use of present tense may imply that something is usually done that way.

Reviewer 3 ·

Basic reporting

The authors have made extensive revision and the article meet my requirement of publishing. The article use clear, unambiguous, technically correct text. The structure, figures. tables meet the journal requirement

Experimental design

The experiment design is kind of regular, perhapes lack of novelty. The method is described in detail

Validity of the findings

The acceptance is based on the new application of texture analysis in drone based remote sensing. So in term of validity of the findings, i recommend this article published.

Additional comments

PeerJ is a new thing to me. If in other computer science related journals, this article can't meet my requirement due to the lack of novelty. However, the PeerJ encourage replication. And in terms of novelty in applications, I recommend it accepted for publication.

---

## Round 0.3 · accepted · Accept

Thank you for making extensive revisions. The revised article meets the requirements of publishing in PeerJ. It is written with clarity using technically correct text. The methodology is also described in detail. The acceptance of this article is based on the new application of texture analysis in drone-based remote sensing.